# Antimicrobial Synergy Testing: Comparing the Tobramycin and Ceftazidime Gradient Diffusion Methodology Used in Assessing Synergy in Cystic Fibrosis-Derived Multidrug-Resistant *Pseudomonas aeruginosa*

**DOI:** 10.3390/antibiotics10080967

**Published:** 2021-08-12

**Authors:** Ijeoma N. Okoliegbe, Karolin Hijazi, Kim Cooper, Corinne Ironside, Ian M. Gould

**Affiliations:** 1Department of Medical Microbiology, Aberdeen Royal Infirmary, Aberdeen AB25 2ZN, UK; kim.cooper@nhs.scot (K.C.); corinne.ironside@nhs.scot (C.I.); ian.gould@nhs.scot (I.M.G.); 2Institute of Dentistry, University of Aberdeen, Aberdeen AB25 2ZR, UK; k.hijazi@abdn.ac.uk

**Keywords:** antimicrobials, ceftazidime, combination antimicrobial susceptibility testing, gradient diffusion, *Pseudomonas aeruginosa*, synergy testing, tobramycin

## Abstract

The need for synergy testing is driven by the necessity to extend the antimicrobial spectrum, reducing drug dosage/toxicity and the development of resistance. Despite the abundance of synergy testing methods, there is the absence of a gold standard and a lack of synergy correlation among methods. The most popular method (checkerboard) is labor-intensive and is not practical for clinical use. Most clinical laboratories use several gradient synergy methods which are quicker/easier to use. This study sought to evaluate three gradient synergy methods (direct overlay, cross, MIC:MIC ratio) with the checkerboard, and compare two interpretative criteria (the fractional inhibitory concentration index (FICI) and susceptibility breakpoint index (SBPI)) regarding these methods. We tested 70 multidrug-resistant *Pseudomonas aeruginosa*, using a tobramycin and ceftazidime combination. The agreement between the checkerboard and gradient methods was 60 to 77% for FICI, while agreements for SBPI that ranged between 67 and 82.86% were statistically significant (*p* ≤ 0.001). High kappa agreements were observed using SBPI (Ƙ > 0.356) compared to FICI (Ƙ < 0.291) criteria, and the MIC:MIC method demonstrated the highest, albeit moderate, intraclass correlation coefficient (ICC = 0.542) estimate. Isolate resistance profiles suggest method-dependent synergism for isolates, with ceftazidime susceptibility after increased exposure. The results show that when interpretative criteria are considered, gradient diffusion (especially MIC:MIC) is a valuable and practical method that can inform the treatment of cystic fibrosis patients who are chronically infected with *P. aeruginosa*.

## 1. Introduction

The emergence of multidrug-resistant (MDR) bacteria is becoming progressively more widespread, especially in cystic fibrosis (CF), where persistent colonization of the lungs leads to prolonged prophylactic treatment, thereby increasing the likelihood of multidrug-resistant organisms [1]. In the CF population, *Pseudomonas aeruginosa* in particular colonizes 70% of CF adults by the age of 25 [2,3]. Therefore, various treatment approaches (such as combination therapy) are employed in patient management to delay the development of resistance to any individual drug during treatment, thus controlling the emergence of multidrug-resistant strains [1,4]. In addition, the use of two anti-pseudomonal agents (for example, β-lactams and aminoglycosides) for combination therapy, where one/both are generally effective, is recommended in the management of infective pulmonary exacerbations, due to their synergistic activity [4,5]. However, there is a lack of evidence guiding the clinician in decisions regarding the best antimicrobial combination that will give a positive outcome [4,6,7], especially in CF chronic *P. aeruginosa* infections, where there is often discordance between susceptibility results and clinical outcome [8]. In the laboratory, there is generally the use of several methods that yield inconsistent results [4], and a lack of accepted standards for in vitro antimicrobial synergy testing. Most methods, such as the checkerboard and time-kill assay, are well-tested methods for evaluating synergy in vitro, but they are labor-/time-intensive and are not scalable as a therapeutic tool in clinical microbiology [9]. To mitigate these issues, the gradient diffusion method has been employed in clinical microbiology for susceptibility/synergy testing [10]. These gradient diffusion plastic strips, impregnated with a pre-defined antibiotic concentration gradient, are placed on a pre-inoculated streaked agar plate, with elliptical inhibition zones as indicators of minimum inhibitory concentration (MIC) at the intersections of the inhibition zone, and the strip edge is read following overnight incubation [10]. This provides a rapid, simple, and accurate method to measure the MIC of antimicrobials and, when adapted for use in combination, is an equally straightforward method for assessing synergy in clinical laboratories [10,11]. In practice, several gradient methods, such as the cross (90° angle), direct overlay (fixed ratio), agar diffusion and MIC:MIC method, have been employed for assessing synergy, but there is currently no consensus on its agreement with any reference standard in assessing synergy [11,12] in *P. aeruginosa*.

The Cystic Fibrosis Susceptibility Testing Service (CFASS) performs extended susceptibility and synergy testing on Gram-negative bacterial isolates recovered from the respiratory tracts of CF adults in Scotland. At present, the service uses the direct overlay method to assess the synergy between two antimicrobial agents. The aim of this study was to compare and correlate three gradient diffusion methods (cross, direct overlay, and the MIC:MIC method) with a reference method (the checkerboard assay). The agar dilution method was not included, as it is deemed too labor-intensive for most clinical laboratories [12]. A secondary objective of this study was to determine the effect of the interpretative criteria on the observed antimicrobial combination results. Our findings are expected to provide empirical evidence on the interpretation of combination testing results employed in the treatment of CF pulmonary exacerbations.

## 2. Results

### 2.1. Study Results

#### 2.1.1. Strain Characteristics

During the study period, 70 MDR *P. aeruginosa* isolates underwent antimicrobial combination testing, using three gradient diffusion methods and the microbroth checkerboard assay. The interpretative category results were determined, using the gradient and checkerboard assay. Of these, 27.14% (n = 19/70) were susceptible to tobramycin, while none of the 70 isolates exhibited ceftazidime MIC values in the susceptible range; however, 37.14% (n = 26/70) were classified as susceptible with increased exposure. The proportion of isolates that were resistant to both tobramycin and ceftazidime was 47.14% (n = 33/70). The average strain MIC value was 33.38 mg/L (MIC_50_ = 3.0 mg/L, MIC_90_ = 48 mg/L) and 28.02 mg/L (MIC_50_ = 4.0 mg/L, MIC_90_ = 64 mg/L) for the tobramycin gradient and checkerboard assay, respectively, while 178.04 mg/L (MIC_50_ = MIC_90_ ≥ 256 mg/L) and 177.39 mg/L (MIC_50_ = MIC_90_ ≥ 256 mg/L) were observed for ceftazidime. 

#### 2.1.2. Essential and Categorical Agreement of Single- and Combination-MIC Testing

The essential and categorical levels of agreements for observed MIC values using the gradient and checkerboard assay were evaluated using Cohen’s kappa Ƙ statistics. Figure 1 shows that the essential agreement of the obtained single-agent MIC values for tobramycin was 91.43% (n = 64/70) within ≤1 two-fold dilution, while 74.29% (n = 52/70) was observed for ceftazidime. Of these, 37.14% (n = 26/70) and 52.86% (n = 37/70) absolute agreement was observed for tobramycin and ceftazidime, respectively. Further analysis for categorical agreement using kappa statistics showed that the levels of agreement between both methods were 88.57% (Ƙ = 0.712, *p* ≤ 0.001) for tobramycin, while 82.86% (Ƙ = 0.559, *p* ≤ 0.001) was observed for ceftazidime. Interestingly, when we analyzed our synergy gradient MIC values for agreement with the checkerboard assay, Figure 1 showed that there was a reduced essential agreement, in the proportions of isolates for combination MICs within ≤1 two-fold dilution, for both tobramycin and ceftazidime. While similar reductions were observed for all tobramycin gradient diffusion methods, the MIC:MIC method demonstrated the least (~20%) reduction compared with the checkerboard assay for ceftazidime.

#### 2.1.3. FICI and SBPI of *P. aeruginosa* Isolate Synergy Testing

The combination MIC values from 70 patient-unique *P. aeruginosa* samples were used in our evaluation of the three gradient synergy methods, with the checkerboard method as a comparator. Using the FICI as interpretative criteria (Figure 2), the checkerboard method produced 28.57% isolates displaying synergy (n = 20) in the study population. Comparison of the gradient synergy population with the comparator showed a decrease for both the MIC:MIC (17.14%, n = 12) and direct overlay (8.57%, n = 6) methods, while 60% increase was observed using the cross method (45.71%, n = 32). In contrast, the comparative analysis of SBPI values (Figure 2) showed an increase in the proportion of isolates for all three gradient diffusion methods. The cross method (8.82%, n = 37) produced the least increase, while direct overlay (47.06%, n = 50) produced the most. 

#### 2.1.4. FICI and SBPI Comparator Agreement

We further analyzed our FICI and SBPI data for agreement with the comparator using Cohen’s kappa statistics. Table 1 showed that, although FICI values had 77.14% (direct overlay), 71.43% (MIC:MIC) and 60% (cross) categorical agreement, poor kappa statistics were observed. Of the 20 checkerboard synergistic isolates, the cross method showed 60% absolute agreement, while 25% and 30% were observed for the direct overlay and MIC:MIC methods, respectively (Table 1). 

In contrast, Table 2 shows that a statistically significant categorical agreement was observed, using the SBPI criteria, for all gradient methods. The highest level of agreement (82.86%) was observed for the cross method (Ƙ = 0.662, *p* < 0.001), while 77.14% was observed for the MIC:MIC method (Ƙ = 0.556, *p* < 0.001). Interestingly, for indices < 2.0, there was ≈88–91% agreement among the three gradient methods. However, marked differences were observed in the SBPI 2–50 range, with the cross (80%) and MIC:MIC (68.57%) methods showing the highest agreement. The lowest level of agreement (45.71%) was shown using the direct overlay method. 

Interestingly, when we used the intraclass correlation coefficient to evaluate the strength of the comparator relationship in method-derived synergy MIC values, moderate ICC estimates of 0.542, with a 95% confidence interval from 0.469 to 0.607 (F(419) = 3.424, *p* < 0.001), were observed using the MIC:MIC method, while the cross and direct overlay methods both produced poor ICC estimates (Table 3).

#### 2.1.5. Effect of Resistance Profiles on FICI and SBPI Values 

Finally, we analyzed our data to understand the effect of resistance profiles on the FICI and SBPI values. Table 4 shows that the cross method overpopulated synergy for all resistant profiles except the tobramycin-susceptible isolates. In contrast, low synergism was observed for those isolates with profiles of ceftazidime susceptibility after increased exposure using the direct overlay and MIC:MIC methods. Indeed, our data showed that, for these isolates, a discordant synergy result (0–67%) was observed for all methods. Interestingly, the SBPI analysis showed that the direct overlay method produced more isolates with low SBPI values for ceftazidime-resistant isolates. In addition, a pairwise comparator comparison (Table 4) showed that, of the synergy observed in 7/33 tobramycin- and ceftazidime-resistant isolates, a 57.14% (n = 4/7) agreement was observed using the direct overlay and cross method, while only 28.57% (n = 2/7) was observed for MIC:MIC. In contrast, the resistance profile did not affect comparator agreement, as high proportions were observed for SBPI values < 2.0.

## 3. Discussion

In CF patient management, there is the continuous emergence of multi-/pan-drug-resistant (MDR/PDR) strains, especially *Pseudomonas aeruginosa*, due to the prolonged and frequent administration of antimicrobial agents that are used in the treatment of pulmonary exacerbations [1,2,4,5]. The emergence of these MDR/PDR phenotypes, resistant to several classes of antimicrobials, impacts the therapeutic options in clinical practice [13]. As a result, in chronic pulmonary exacerbations management, physicians are compelled to employ combination therapy using ≥2 antibiotics, of which the inhibitory effects of the antibiotics used together are greater than the sum of each agent’s activity in monotherapy [14]. There is a dearth of evidence on the best antimicrobial combination that will give a positive outcome [4,6,7,13] in clinical management, as the response of *P. aeruginosa* to various antimicrobial combinations has been shown to be unpredictable [11]. This is further confounded by the lack of in vitro synergy testing standardization, and the absence of gold standards. Kidd et al. [14] reported that disk and gradient diffusion tests are mostly used in clinical microbiology laboratories, due to their low cost and ease of performance, while dilution tests are commonly employed in research and reference laboratory settings. In this study, we compared the performance of three gradient diffusion tests for synergy testing to the checkerboard micro broth dilution method. We chose ceftazidime and tobramycin combinations as the evaluated antimicrobial combinations, as this treatment is commonly used in the management of CF-MDR *P. aeruginosa* pulmonary exacerbations [5] and has been previously described as producing synergistic interactions [15].

In this study, as described in previous studies [12,16], our data suggest that the observed single-agent gradient MIC results correlated well (>82% categorical agreement) with those values obtained using the checkerboard assay. Lasko et al. [16] recently demonstrated that at least an 80% categorical agreement was observed in a comparative analysis of the gradient method vs. the broth microdilution assays for ceftazidime, ceftolozane/tazobactam, meropenem and tobramycin in CF-derived *P. aeruginosa*. Similarly, in agreement with Pankey et al. [12], the median single-agent MIC values were at ≤1 two-fold dilution for both methods and antimicrobials. These levels of agreement and the ease of performance lend credence to the potential of gradient diffusion methods for evaluating in vitro antimicrobial synergy. There is currently no clear consensus on the gold standard for method evaluation, and no agreement exists on the best method. Several methods have been described and evaluated, for agreement with checkerboard or time-kill assays, as reference standards using numerous bacteria/antimicrobial combinations [13]. The time-kill method has been shown to yield high concordance in various studies, but because it produces dynamic and longitudinal information that cannot be obtained by other methods, its use as the gold standard for synergy testing is debatable [17]. In contrast, the broth microdilution assay and the gradient synergy method measure the inhibitory activities of antibiotics [12]. Therefore, we used the checkerboard assay to assess the levels of synergy observed for each gradient method. Mirrored in our data, as described by other synergy studies [12,18], is the observation that the gradient direct overlay method tends to show the effects of the most active agent, rather than the interactions between both agents, and often underestimates synergism. However, unlike Balke et al. [18], we observed lower synergy proportions; a plausible explanation of the difference might be the study isolate population, since their study was composed of 71% tobramycin-susceptible strains, thus increasing the synergy rates. Indeed, we demonstrate a method-dependent synergism using isolates with resistance profiles of ceftazidime susceptibility after increased exposure. In addition, our data (Appendix A) reveals statistically significant differences when tobramycin gradient strips were placed before ceftazidime strips, suggesting that an antibiotic-dependent effect might be responsible for the inconsistencies. We propose that further research should be explored to enrich our understanding of antibiotics synergy testing. A plethora of evidence suggests several mechanisms of resistance, such as the presence of mobile plasmid-borne β-lactamase genes, as well as changes in the cell outer membrane through the decrease in the number of porins present and mutations that change the selectivity of porin channels [19,20]. Other mechanisms affecting drug uptake are the possession of chromosomal-encoded genes for efflux pumps that can be expressed (constitutively induced) or overexpressed modifications in the 30S and 50S ribosomal subunits [19,20]. Although not a remit of this study, the molecular characterization of study isolates would enrich our knowledge on the impact of resistant markers on antimicrobial uptake during synergy testing.

There is growing interest in understanding the unpredictability of synergy results. Gómara et al. [13] reported that, depending on the mathematical model, drug interactions can lead to opposite conclusions with the lack of translation, due to the use of MIC as the basis for calculation. Hence, the standard reference parameter (FICI) for quantification of pairwise drug interactions was modified to more restrictive criteria, where synergy was described as a ≥four-fold reduction in the MICs of both compounds when in combination, compared with their MICs alone [13,21]. Despite these restrictive criteria, there is still an absence of consensus and standardization in synergy studies. Hence, our lab proposed the use of SBPI as a useful parameter for comparing the in vitro effectiveness of antimicrobial combinations. Milne et al. [15] proposed that this parameter, defined as the summation of inverse proportions of the organisms’ susceptible breakpoint and the combination’s MIC, might be useful in assessing the outcome of antimicrobial combination testing. Indeed, our results present strong in vitro evidence that SBPI has the potential to remove the unpredictability associated with the FICI criteria, notwithstanding the need for outcome studies to evaluate the efficacy of synergy testing in the CF population. However, Aaron et al. [6], in a 4.5-year, randomized, double-blind controlled study involving 132 patients, concluded that there was no difference in the lung function, dyspnea, bacterial density or treatment failure in the study cohort of patients who were treated with the multiple-combination bacterial antimicrobial testing. While results from this study are disappointing because of the disagreements between different methods and the uncertainty of the FICI criteria, we propose that a holistic approach involving several synergy methods and interpretative criteria should be explored in clinical trials, to unravel synergy testing. 

Limitations of this study include the choice of microorganism and/or antimicrobial agents, single-rater analyses, and test concentrations and the use of the highest doubling dilution of antimicrobials above those used in clinical practice. 

Although it is difficult to conclusively demonstrate the absolute advantages of each synergy method/interpretative criterion, this research reiterates the need for clinical microbiology laboratories to standardize synergy testing, and for setting up a single clear definition for synergy/clinical relevance. It also gives empirical in vitro evidence that SBPI should be further explored as the interpretative criterion, due to the uncertainties associated with the FICI criterion. Finally, due to synergy under- and overpopulation using the direct overlay and cross methods, respectively, the results suggest the adoption of the MIC:MIC as the preferred method for studying antibiotic interactions in diagnostic samples.

## 4. Materials and Methods

### 4.1. Study Isolates and Media 

Of the 721 MDR *P. aeruginosa* sent to CFASS for extended antimicrobial susceptibility testing over a 20-year period (2001–2020), one-tenth of the patient-unique isolates were randomly selected for this study. These isolates, stored in the bacterial preservation system MICROBANKTM (Pro-Lab Diagnostics, Richmond Hill, ON, Canada) at −80 °C, were plated onto Mueller-Hinton (MH) agar (Oxoid Ltd., Basingstoke, UK) and were identified using a Vitek^®^ MS (BioMérieux, Inc., Durham, NC, USA). Isolates were tested by CFASS and were classified as multidrug-resistant if there was acquired non-susceptibility to at least one agent in the ≥3 antimicrobial group [22]. The media used for minimum inhibitory concentration (MIC) and synergy testing included Mueller-Hinton agar and cation-adjusted Mueller-Hinton II broth (CAMHB). *P. aeruginosa* ATCC 27853 was used as a quality control strain to validate the MIC values.

### 4.2. Antimicrobial Agents 

Standard laboratory powders of an aminoglycoside, tobramycin (TM) and a β-lactam, ceftazidime (TZ) hydrate (both manufactured by Sigma-Aldrich, St Louis, MO, USA) were used for the checkerboard experiments. For the gradient diffusion experiments, test strips (BioMérieux, Basingstoke, UK) of both antimicrobial agents were used, according to the manufacturer’s instructions.

### 4.3. Gradient Diffusion MIC Testing

The MICs for each isolate were determined, using TM and TZ gradient test strips. Briefly, a saline suspension of 0.5 McFarland standard (1.0 for mucoid strains) from 24-hour cultures was inoculated onto MH agar, according to the EUCAST guidelines for disk diffusion plate inoculation [15]. Two gradient test strips (TM and TZ) were placed top-to-tail, according to the manufacturer’s instructions. These plates were incubated in ambient air at 35 °C for 18 ± 2 h. After incubation, the MIC values were read at ellipse intersections of the MIC reading scale. MICs were interpreted as susceptible (S), susceptible with increased exposure (I) (intermediate resistance), or resistant (R), based on the guidelines published by European Clinical Antimicrobial Susceptibility Testing [23]. MIC values for all isolates were obtained as the mean MIC of three independent biological replicates. Categorical agreement was defined as an agreement of interpretative results (SIR) with the comparator, while essential agreement was defined as an agreement within ±1 two-fold dilution with the reference standard [24,25]. The MICs of the antimicrobials that inhibited 50% and 90% of the isolates were calculated and expressed as MIC_50_ and MIC_90,_ respectively.

### 4.4. Gradient Diffusion Synergy Methods 

Assessment of the synergy methods was carried out for each method using three biological replicates, according to the methods employed by Pankey et al. [12]. As described, bacterial suspensions of each isolate were prepared in a saline solution adjusted to 0.5 McFarland standard, and an even lawn was spread onto MH agar. The placement of the gradient test strip was different for each method. The combination MIC was read after 18 ± 2 h incubation in ambient air at 35 °C and interpreted as the value at which the inhibition zone intersected the scale on each respective gradient diffusion strip.

#### 4.4.1. Cross Method

Test strips for antimicrobials TM and TZ were placed on the inoculated 90 mm MH agar plate in a cross formation, with a 90° angle at the intersection between the scales at their previously determined respective MICs. 

#### 4.4.2. Direct Overlay Method

TM and TZ test strips were placed top-to-tail on different sections of a 90-mm MH agar and incubated at room temperature. After 1 h, to allow for antimicrobial diffusion into the agar, the test strips were removed and discarded. A fresh test strip of the opposite antimicrobial was placed directly over the imprint of the first antimicrobial (i.e., the TM strip was replaced with a fresh TZ strip, and vice versa). 

#### 4.4.3. MIC:MIC Overlay Method

TM and TZ strips were placed on different sections of a 150-mm MH agar plate. The agar was marked with an inoculating loop adjacent to the previously determined MIC value for each strip. For isolates where the MIC exceeded the concentration on the test strip, the highest concentration was marked on the agar. The strips were removed and discarded after 1 h incubation at room temperature, to enable the antimicrobials to diffuse into the agar. A fresh opposite test strip was placed over the imprint of the discarded strip so that the MIC of the fresh strip lined up with the mark that signified the MIC of the discarded strip. 

### 4.5. Broth Microdilution Checkerboard Method

To validate the synergy results, checkerboard assays were performed on the isolates according to Leber et al. [26]. Briefly, 24-hour isolates were inoculated into CAMHB, and incubated at 35 °C until the exponential growth phase. Using a 96-well plate, a two-fold dilution of freshly prepared antimicrobial at different concentrations was dispensed in a checkerboard array and inoculated with 10^5^ CFU/ml of bacterial suspension. One well with no antibiotics was used as the positive control. After incubation at 35 °C, plates were examined for visual turbidity using the magnifying mirror reader; growth was indicated as turbidity in wells. The MIC value was derived as the lowest drug concentration with no visible growth [27,28]. MIC values for all isolates were obtained as the mean MIC of three independent biological replicates.

### 4.6. Interpretative Criteria

#### 4.6.1. Fractional Inhibitory Concentration Index (FICI)

The synergy MIC was expressed using the FICI and calculated as described below.
FICI = (MIC A combination/MIC A single) + (MIC B combination/MIC B single)

If the MIC value was greater than the antimicrobial range tested, the next doubling dilution above this value was used to calculate the FICI (e.g., if an MIC of >32 mg/L was found, then the FICI was calculated using 64 mg/L) [29]. These indices were interpreted as synergy (FICI ≤ 0.5), no interaction (FICI > 0.5 and ≤4.0) and antagonism (FICI > 4.0) [21].

#### 4.6.2. Susceptible Breakpoint Index (SBPI)

The SBPI was used to describe synergy analysis and calculated as described below.

SBPI = (susceptible breakpoint of antimicrobial A/MIC of antimicrobial A combination) + (susceptible breakpoint of antimicrobial B/MIC of antimicrobial B combination) [15]. These combination results were categorized in rank order, according to their decreasing SBPI results. Values were interpreted as poor for values < 2, good for 2–50, and very good for 50–100, while >100 was exceptional. 

### 4.7. Statistical Analysis

Statistical analyses of the categorical and continuous variables were carried out using Microsoft Office Excel 2013 and IBM SPSS Statistics for Windows, Version 25 (IBM Corp., Armonk, NY, USA). Categorical agreement of FICI and SBPI values was determined using Cohen’s kappa (Ƙ) statistics and was interpreted as poor for values that were Ƙ < 0.40, and very good for Ƙ > 0.75, while values between 0.40 and 0.75 were classified as good [12]. A *p*-value of <0.05 was considered statistically significant. All numerical gradient-derived synergy MIC values were used to calculate the intraclass correlation coefficient (ICC) estimates and their 95% confident intervals, using an SPSS based on a single-rating, absolute agreement, two-way mixed-effects model. ICC values of less than 0.5 were indicative of poor reliability, 0.5–0.75, moderate reliability, and 0.75–0.90, good reliability, while values greater than 0.90 were indicative of excellent reliability [30].

## Figures and Tables

**Figure 1 antibiotics-10-00967-f001:**
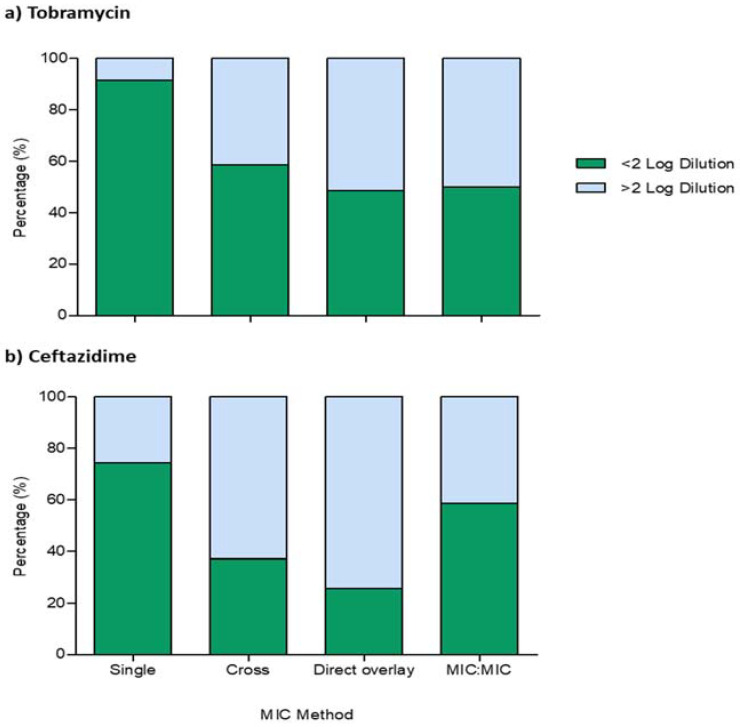
Proportions of isolates with an essential agreement to the comparator. Measurements are expressed as relative log differences in gradient MIC values, compared with the checkerboard assay, for (**a**) tobramycin and (**b**) ceftazidime; green bars represent the proportion of isolates with ≤2 log dilutions, while isolates ≥ 2 log dilutions are represented with blue bars. All experiments were carried out as triplicates.

**Figure 2 antibiotics-10-00967-f002:**
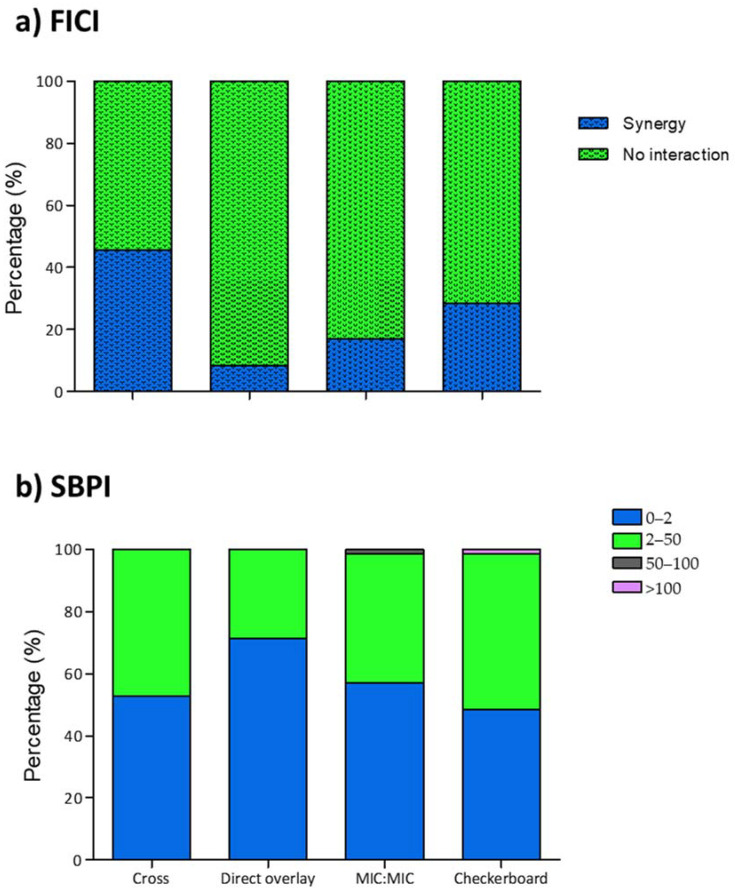
Observed FICI and SBPI proportions in study isolates. Measurements are expressed as proportions of synergy for (**a**) FICI and calculated values for (**b**) SBPI, using each MIC method. Blue bars represent the proportion of isolates with synergistic (dotted) and poor SBPI values, while no interaction (dotted) and good SBPI values are represented with green bars. All experiments were carried out as triplicates.

**Table 1 antibiotics-10-00967-t001:** Concordance among the checkerboard and the three gradient diffusion methods, using the FICI criteria (n = 70).

	Checkerboard			Categorical	Kappa Statistics	
Gradient Method	Synergy	Indifference	Total	Agreement (%)	*K*	*p*-Value
**Cross**				60.00	0.169	0.129
Synergy	12	20	32			
Indifference	8	30	38			
**Direct overlay**				77.14	0.291	0.002
Synergy	5	1	6			
Indifference	15	49	64			
**MIC:MIC**				71.43	0.205	0.071
Synergy	6	6	12			
Indifference	14	44	58			

**Table 2 antibiotics-10-00967-t002:** Concordance among the checkerboard and the three gradient diffusion methods, using the SBPI criteria (n = 70).

	Checkerboard					Categorical	Kappa Statistics	
Gradient Method	0–2	2–50	50–100	>100	Total	Agreement (%)	*K*	*p*-Value
**Cross**						82.86	0.662	<0.001
0–2	30	7	0	0	37			
2–50	4	28	0	1	33			
50–100	0	0	0	0	0			
>100	0	0	0	0	0			
**Direct overlay**						67.14	0.356	0.001
0–2	31	19	0	0	50			
2–50	3	16	0	1	20			
50–100	0	0	0	0	0			
>100	0	0	0	0	0			
**MIC:MIC**						77.14	0.556	<0.001
0–2	30	10	0	0	40			
2–50	4	24	0	1	29			
50–100	0	1	0	0	1			
>100	0	0	0	0	0			

**Table 3 antibiotics-10-00967-t003:** Intraclass coefficient analysis of gradient diffusion methods, using an absolute agreement, two-way mixed model.

	Intraclass	95% Confidence Interval		F Test with True Value 0			
Method	Correlation	Lower Bound	Upper Bound	Value	*df*1	*df*2	*p*-Value
Cross	0.386	0.302	0.465	2.280	419	419	<0.001
Direct Overlay	0.259	0.165	0.347	1.743	419	419	<0.001
MIC:MIC	0.542	0.469	0.607	3.424	419	419	<0.001

**Table 4 antibiotics-10-00967-t004:** Effect of resistance profile on FICI and SBPI values in the study population.

					Gradient Diffusion Method % ^a^ (No. of Isolates) ^b^				Comparator Agreement % ^a^ (No. of Isolates) ^b^		
Resistance Profile		No *	Index	Result	CB ^£^	Cross	DO ^£^	MIC:MIC	Cross	DO ^£^	MIC:MIC
TM (R ^#^)	TZ (R ^#^)	33	FICI	Syn ^$^	21.21 (7)	33.33 (11)	15.15 (5)	18.18 (6)	57.14 (4)	57.14 (4)	28.57 (2)
				NI ^$^	78.79 (26)	66.67 (22)	84.85 (28)	81.82 (27)	69.23 (18)	96.15 (25)	84.62 (22)
			SBPI	≤2.0	63.64 (21)	78.79 (26)	96.97 (32)	72.73 (24)	95.24 (20)	100 (21)	90.48 (19)
				2.0–50.00	36.36 (12)	21.21 (7)	3.03 (1)	24.24 (8)	50.00 (6)	8.33 (1)	50.00 (6)
TM (R ^#^)	TZ (I ^#^)	18	FICI	Syn ^$^	27.78 (5)	66.67 (12)	5.56 (1)	5.56 (1)	100.00 (5)	20.00 (1)	20.00 (1)
				NI ^$^	72.22 (13)	33.33 (6)	94.44 (17)	94.44 (17)	46.15 (13)	100.00 (13)	100.00 (13)
			SBPI	≤2.0	66.67 (12)	55.56 (10)	61.11 (11)	83.33 (15)	83.33 (10)	75.00 (9)	91.67 (11)
				2.0–50.00	33.33 (6)	44.44 (8)	38.89 (7)	16.67 (3)	100.00 (6)	66.67 (4)	33.33 (2)
TM (S ^#^)	TZ (R ^#^)	11	FICI	Syn ^$^	45.45 (5)	36.36 (4)	0(0)	45.45 (5)	20.00 (1)	0(0)	60.00 (3)
				NI ^$^	54.55 (6)	63.64 (7)	100.00 (11)	54.55 (6)	50.00 (3)	100.00 (6)	66.67 (4)
			SBPI	≤2.0	9.09 (1)	0(0)	45.45 (5)	0(0)	0(0)	100.00 (1)	0(0)
				2.0–50.00	81.82 (9)	100.00 (11)	54.55 (6)	100.00 (11)	100.00 (9)	55.56 (5)	100.00 (9)
TM (S ^#^)	TZ (I ^#^)	8	FICI	Syn ^$^	37.50 (3)	62.50 (5)	0(0)	0(0)	66.67 (2)	0(0)	0(0)
				NI ^$^	62.50 (5)	37.50 (3)	100.00 (8)	100.00 (8)	40.00 (2)	100.00 (5)	100.00 (5)
			SBPI	≤2.0	0(0)	12.50 (1)	25.00 (2)	12.50 (1)	0(0)	0(0)	0(0)
				2.0–50.00	100.00 (8)	87.50 (7)	75.00 (6)	87.50 (7)	87.50 (7)	75.00 (6)	87.50 (7)

^#^ M, tobramycin; TZ, ceftazidime; R, resistant; I, susceptible after increased exposure; S, susceptible; * total number of isolates in the resistance profile; ^$^ Syn, synergistic; NI, no interaction; ^£^ CB, checkerboard; DO, direct overlay; ^a^ proportion of isolate; ^b^ number of isolates.

## Data Availability

All data presented in this study are available on request by contacting the corresponding author.

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
