# Peer review of "Antimicrobial Synergy Testing: Comparing the Tobramycin and Ceftazidime Gradient Diffusion Methodology Used in Assessing Synergy in Cystic Fibrosis-Derived Multidrug-Resistant Pseudomonas aeruginosa"

_antibiotics, 2021, doi:10.3390/antibiotics10080967_

Round 1
Reviewer 1 Report
The authors compared different MIC and synergy testing methods, and suggested that the gradient diffusion (esp. MIC:MIC) was a valuable and practical method to assess the combination synergy. However, to conclude which method could be preferred in assessing the synergy in P. aeruginosa, it is better to use more combination treatments other than only tobramycin-ceftazidime combination, the rational for choosing this two antibiotics is to be elaborated.
Author Response
Dear Reviewer,
Thank you for the consideration of our manuscript entitled, “Antimicrobial synergy testing: comparing the tobramycin and ceftazidime gradient diffusion methodology used in assessing synergy in cystic fibrosis derived multidrug-resistant Pseudomonas aeruginosa,” with manuscript number Antibiotics-1301661. We found your critique of our initial submission to be very helpful and have revised the manuscript. In responding to your comments and suggestions, we believe our manuscript has been greatly strengthened and represents a significant improvement over our initial submission. Our point-by point responses of the critique are outlined below;
Reviewer 1
- The authors compared different MIC and synergy testing methods, and suggested that the gradient diffusion (esp. MIC:MIC) was a valuable and practical method to assess the combination synergy. However, to conclude which method could be preferred in assessing the synergy in P. aeruginosa, it is better to use more combination treatments other than only tobramycin-ceftazidime combination, the rational for choosing this two antibiotics is to be elaborated.
The authors would like to agree with the reviewer that it would have made our research richer if we considered other antimicrobial combination treatments other than only tobramycin-ceftazidime combination in our study. First, in clinical practice, the use of two anti-pseudomonal agents for combination therapy of which one/both are generally effective is recommended in the management of infective pulmonary exacerbations due to their synergistic activity (Brennan-Krohn et. al., 2019, Stefani et al., 2017). Furthermore, Bassetti et al. 2019, further proposed the administration of this combination when P. aeruginosa is suspected to balance between early antibiotics administration and the risk of resistance selection while previous work published by our lab show synergistic activities with this combination. Finally, observations from Pankey et al., 2013 using meropenem and colistin in Klebsiella pneumoniae suggest a non-antibiotic/species effect
Thank you again for your invaluable feedback and the opportunity to review our manuscript.
Sincerely,
Ijeoma Okoliegbe, PhD
Institute of Dentistry
University of Aberdeen,
Aberdeen, UK,
AB25 2ZD
Tel.: +44 (0) 1224-437532
Email: [email protected]

Reviewer 2 Report
The aim of the study was to investigate a very interesting issue of antimicrobial synergy testing. I suppose a lot of effort went into the study. Novelty, a significant impact of the obtained results as well as several findings made on the basis of them, in my opinion, are really interesting for the readers.
My major concerns are:
- What is the explanation/reference for 1 McFarland suspension application for mucoid strains?
- Why the origin (cystic fibrosis patients) of MDR aeruginosa strains is so underlined and discussed? Are this particular strains different than other MDR isolates derived from other clinical specimens?
- The description of the applied methodology, in my opinion, is really superficial. It should be illustrated with some pictures since it is rather sophisticated for a particular reader and not explained sufficiently,
- Tables and Figures, in my opinion, should be prepared and described in a different/better way to make it easier to understand without referring to the text,
- The addition of Figures (at least as supplementary material), showing the results of the applied methods would be useful and would make the M&M section easier to understand,
- The addition of Formulas for calculations of FICI and SBPI (with particular examples from the manuscript) would make their application easier to understand,
- There is lack of information if the same strains, in each repeat of the tests, presented the same result of susceptibility results/interpretation,
- Since some of the experiments were carried out in triplicates, it should be noted somehow, what differences were observed between the mentioned repeats,
My minor concerns are:
- There are some typing errors, e.g -80oC, 35oC or 105 CFU/ml that should be corrected before the publication,
- Keywords should be listed in an alphabetical order, in my opinion,
- There is plenty of square brackets in the text, making the sense of the sentences difficult to understand what confuse the reader,
- Lack of italics in some places where should be used, e.g. in vitro, P. aeruginosa, etc.
However, all the points mentioned above do not decrease the overall value of the research.
Author Response
26th July, 2021
Dear Reviewer,
Thank you for the consideration of our manuscript entitled, “Antimicrobial synergy testing: comparing the tobramycin and ceftazidime gradient diffusion methodology used in assessing synergy in cystic fibrosis derived multidrug-resistant Pseudomonas aeruginosa,” with manuscript number Antibiotics-1301661. We found your critique of our initial submission to be very helpful and have revised the manuscript. In responding to your comments and suggestions, we believe our manuscript has been greatly strengthened and represents a significant improvement over our initial submission. Our point-by point responses of the critique are outlined below;
Reviewer 2
The aim of the study was to investigate a very interesting issue of antimicrobial synergy testing. I suppose a lot of effort went into the study. Novelty, a significant impact of the obtained results as well as several findings made on the basis of them, in my opinion, are really interesting for the readers.
Major concerns
- What is the explanation/reference for 1 McFarland suspension application for mucoid strains?
The authors would like to thank the reviewer for this comment and have now added a reference for the use of 1 McFarland suspension for mucoid strains
- Why the origin (cystic fibrosis patients) of MDR aeruginosastrains is so underlined and discussed? Are this particular strains different than other MDR isolates derived from other clinical specimens?
The authors would like to respond that our discussion to define this patient population is due to our CF experience rather than differences in the susceptibility patterns. We do not have any data to establish a difference as this was beyond the scope of our work.
- The description of the applied methodology, in my opinion, is really superficial. It should be illustrated with some pictures since it is rather sophisticated for a particular reader and not explained sufficiently
The authors would like to thank the reviewer and have presented a visual illustration of the applied methodology as a supplementary file.
- Tables and Figures, in my opinion, should be prepared and described in a different/better way to make it easier to understand without referring to the text
The authors would like to thank the reviewer for the comment and have now edited the figures.
- The addition of Figures (at least as supplementary material), showing the results of the applied methods would be useful and would make the M&M section easier to understand
The authors express gratitude for this comment and have included the addition of figures as a supplementary material for easy comprehension of the materials and methods.
- The addition of Formulas for calculations of FICI and SBPI (with particular examples from the manuscript) would make their application easier to understand
The authors would respond by stating that specific examples on how to calculate the FICI and SBPI have now been added as a supplementary file
- There is lack of information if the same strains, in each repeat of the tests, presented the same result of susceptibility results/interpretation
The authors express gratitude for this comment and have now included the testing data as a supplemental file.
- Since some of the experiments were carried out in triplicates, it should be noted somehow, what differences were observed between the mentioned repeat.
The authors would respond by expressing gratitude for this observation. We have now included the supplemental file which displays the differences observed using each method. Though it would have enriched our study, statistical analysis of repeatability was not a remit of our single rater study.
My minor concerns are:
- a) There are some typing errors, e.g -80oC, 35oC or 105 CFU/ml that should be corrected before the publication
The authors are grateful to the reviewer for noticing this error and have now corrected the typing errors.
- b) Keywords should be listed in an alphabetical order, in my opinion
The authors would respond by stating that Keywords have now been alphabetized as advised.
- c) There is plenty of square brackets in the text, making the sense of the sentences difficult to understand what confuse the reader
The authors thank the reviewer for this comment and have made the change
- d) Lack of italics in some places where should be used, e.g. in vitro, P. aeruginosa, etc.
The authors are grateful to the reviewer for noticing this error and have now italicized the words.
However, all the points mentioned above do not decrease the overall value of the research.
Thank you again for your invaluable feedback and the opportunity to review our manuscript.
Sincerely,
Ijeoma Okoliegbe, PhD
Institute of Dentistry
University of Aberdeen,
Aberdeen, UK,
AB25 2ZD
Tel.: +44 (0) 1224-437532
Email: [email protected]
Reviewer 3 Report
Major criticisms:
- Line 227: data not shown on differences in susceptibility profiles due to order of antibiotic administration in gradient dilution assays would be compelling data to present/address. This could be a potential reason for in vitro vs clinical outcome discrepancy and therefore an important avenue of research in the context of antibiotic synergy.
- Line 266: authors claim that the data support SBPI as a promising avenue for in vitro assay results interpretation. SBPI uses arbitrary resistance thresholds for bacterial species, so while this statistic might produce more consistent results for data interpretation, its use as a benchmark might inadvertently minimize the naturally broad variance in Pseudomonas aeruginosa clinical isolate antibiotic resistance, which is an important aspect to consider when making decisions about clinical treatment (arguably, accuracy > precision in this case)
- Line 349: use of next highest doubling dilution when isolate MIC is beyond assay range is subjective/bias and weakens the argument for authors’ claims of meaningful correlations
Minor criticisms:
- Line 23: should read “ceftazidime-susceptibility” not “ceftazidime-susceptible”
- Line 81: please define how MIC50 and MIC90 are calculated
- Line 189: “There…” should be capitalized.
- Line 212: “exist” should read “exists”
- Line 224: authors mention isolate population differences could account for discrepancy between parallel studies – please include supplemental data on raw MICs, resistance profiles, epidemiological data for study isolates (or directory information if publicly available)
- Line 226-227: edit for clarity “…we demonstrate a method-dependent synergism with exposure-induced ceftazidime susceptibility.”
- Line 230-235: not just chromosomal changes! mobile plasmid-born beta-lactamase genes are a major cause of multidrug resistance should be mentioned here
- Line 299: please define “susceptible increased exposure [I]” (for some laboratories a designation of “I” means intermediate resistance)
Author Response
26th July, 2021
Dear Reviewer,
Thank you for the consideration of our manuscript entitled, “Antimicrobial synergy testing: comparing the tobramycin and ceftazidime gradient diffusion methodology used in assessing synergy in cystic fibrosis derived multidrug-resistant Pseudomonas aeruginosa,” with manuscript number Antibiotics-1301661. We found your critique of our initial submission to be very helpful and have revised the manuscript. In responding to your comments and suggestions, we believe our manuscript has been greatly strengthened and represents a significant improvement over our initial submission. Our point-by point responses of the critique are outlined below;
Reviewer 3
Major criticisms:
1) Line 227: data not shown on differences in susceptibility profiles due to order of antibiotic administration in gradient dilution assays would be compelling data to present/address. This could be a potential reason for in vitro vs clinical outcome discrepancy and therefore an important avenue of research in the context of antibiotic synergy.
The authors would like to thank the reviewer for this comment and have now included the data in the supplemental data.
2) Line 266: authors claim that the data support SBPI as a promising avenue for in vitro assay results interpretation. SBPI uses arbitrary resistance thresholds for bacterial species, so while this statistic might produce more consistent results for data interpretation, its use as a benchmark might inadvertently minimize the naturally broad variance in Pseudomonas aeruginosa clinical isolate antibiotic resistance, which is an important aspect to consider when making decisions about clinical treatment (arguably, accuracy > precision in this case).
The authors agree with the reviewer that SBPI is an arbitrary resistance threshold for bacterial species and the natural broad variance of Pseudomonas aeruginosa clinical isolates should also be considered when making decisions about clinical treatment. To mitigate the use of an arbitrary benchmark, the authors have proffered that a clinical trial involving a holistic approach involving several synergy methods and interpretative criteria should be explored to enrich the understanding of synergy testing in the management of multidrug resistant organisms.
3) Line 349: use of next highest doubling dilution when isolate MIC is beyond assay range is subjective/bias and weakens the argument for authors’ claims of meaningful correlations
The authors are grateful for this comment and agree that this might have biased our observations and have now included it as a limitation of the study.
Minor criticisms:
- a) Line 23: should read “ceftazidime-susceptibility”not “ceftazidime-susceptible”
The authors would like to thank the reviewer for noticing this term and have now replaced all ceftazidime-susceptible with ceftazidime-susceptibility.
- b) Line 81: please define how MIC50 and MIC90are calculated
The authors are grateful for this comment and have now defined MIC50 and MIC90
- c) Line 189: “There…” should be capitalized.
The authors are grateful to the reviewer for noticing this typing error which has now been changed.
- d) Line 212: “exist” should read “exists”
The authors are express gratitude for this comment and error has been corrected.
- e) Line 224: authors mention isolate population differences could account for discrepancy between parallel studies – please include supplemental data on raw MICs, resistance profiles, epidemiological data for study isolates (or directory information if publicly available)
The authors would like to thank the reviewer and have presented a supplemental file containing the raw study data
- f) Line 226-227: edit for clarity “…we demonstrate a method-dependent synergism with exposure-induced ceftazidime susceptibility.”
The authors would like to thank the reviewer for noticing this term and have now replaced “ceftazidime susceptible” with “ceftazidime-susceptibility”.
- g) Line 230-235: not just chromosomal changes! mobile plasmid-born beta-lactamase genes are a major cause of multidrug resistance should be mentioned here
The authors are express gratitude for this comment and have now included the presence of mobile plasmid-borne β-lactamase genes.
- h) Line 299: please define “susceptible increased exposure [I]” (for some laboratories a designation of “I” means intermediate resistance)
The authors would like to thank the reviewer for noticing the use of this new EUCAST terminology in our manuscript (v_11.0_Breakpoint_Tables.pdf (eucast.org)) and have now included the old term intermediate resistance for ease of comprehension.
Thank you again for your invaluable feedback and the opportunity to review our manuscript.
Sincerely,
Ijeoma Okoliegbe, PhD
Institute of Dentistry
University of Aberdeen,
Aberdeen, UK,
AB25 2ZD
Tel.: +44 (0) 1224-437532
Email: [email protected]
